# Intensive Care Management of Severe Hyponatraemia—An Observational Study

**DOI:** 10.3390/medicina60091412

**Published:** 2024-08-29

**Authors:** Thomas Roe, Mark Brown, Adam J. R. Watson, Bianca-Atena Panait, Nachiket Potdar, Amn Sadik, Shiv Vohra, David Haydock, Ryan Beecham, Ahilanandan Dushianthan

**Affiliations:** 1General Intensive Care Unit, University Hospital Southampton NHS Foundation Trust, Tremona Road, Southampton SO16 6YD, UK; tom.roe6@outlook.com (T.R.); nasifsadik2000@gmail.com (A.S.);; 2Perioperative and Critical Care Theme, NIHR Southampton Biomedical Research Centre, University Hospital Southampton, University of Southampton, Southampton SO17 1BJ, UK; 3Faculty of Medicine, University of Southampton, Southampton SO17 1BJ, UK

**Keywords:** hyponatraemia, intensive care, overcorrection, osmotic demyelination syndrome

## Abstract

*Background and Subject:* Hyponatraemia is a common electrolyte disorder. For patients with severe hyponatraemia, intensive care unit (ICU) admission may be required. This will enable close monitoring and allow safe management of sodium levels effectively. While severe hyponatraemia may be associated with significant symptoms, rapid overcorrection of hyponatraemia can lead to complications. We aimed to describe the management and outcomes of severe hyponatraemia in our ICU and identify risk factors for overcorrection. *Materials and Methods:* This was a retrospective single-centre cohort that included consecutive adults admitted to the ICU with serum sodium < 120 mmol/L between 1 January 2017 and 8 March 2023. Anonymised data were collected from electronic records. We included 181 patients (median age 67 years, 51% male). *Results:* Median admission serum sodium was 113 mmol/L (IQR: 108–117), with an average rate of improvement over the first 48 h of 10 mmol/L/day (IQR: 5–15 mmol/L). A total of 62 patients (34%) met the criteria for overcorrection at 48 h, and they were younger, presented with severe symptoms (seizures/arrythmias), and had lower admission sodium concentration. They were more likely to be treated with hypertonic saline infusions. Lower admission sodium was an independent risk factor for overcorrection within 48 h, whereas the presence of liver cirrhosis and fluid restriction was associated with normal correction. No difference was identified between the normal and overcorrected cohorts for ICU/hospital length of stay or mortality. *Conclusions:* In some patients with severe hyponatraemia, overcorrection is inevitable to avoid symptoms such as seizures and arrhythmias, and consequently, we highlight the key factors associated with overcorrection. Overall, we identified that overcorrection was common and concordant with the current literature.

## 1. Introduction

Hyponatraemia, characterised by a serum sodium concentration below 135 mmol/L, is a common electrolyte disturbance encountered in the intensive care unit (ICU) setting [1]. This condition can arise from various aetiologies, each presenting unique challenges in management and potential complications. Similarly, severe hyponatraemia (<120 mmol/L) can result from several factors, including (but not exclusively) fluid overload, syndrome of inappropriate antidiuretic hormone secretion (SIADH), adrenal insufficiency, diuretic use, cardiac/liver failure, and renal dysfunction [2]. There may be serious consequences of severe hyponatraemia, leading to altered mental status, seizures, coma, prolonged ICU length of stay, prolonged mechanical ventilation, and increased mortality [3,4].

The management of severe hyponatraemia involves careful assessment of the underlying cause and correction of sodium levels at a controlled rate to avoid osmotic demyelination syndrome (ODS), a rare consequence of rapid overcorrection that manifests as irreversible neurological damage [5]. Therapeutic strategies, according to internationally accepted guidelines, are variable and may include fluid restriction, diuretic therapy, and hypertonic saline in severe symptomatic cases, with close monitoring of serum sodium levels and neurological status [5,6,7]. Management of overcorrection can be achieved with the use of desmopressin (with or without enteral water intake or hypotonic intravenous fluid infusion); however, the exact impact of such a treatment is not entirely predictable, and current guidelines recommend seeking specialist endocrine input before initiating this treatment [5].

Overcorrection of hyponatraemia, according to European guidelines, is defined as a rise in serum sodium concentration exceeding 10 mmol/L within 24 h, more than 18 mmol/L within the first 48 h, or more than 8 mmol/L/day after the initial 24 h of admission [5,8]. Rapid correction of severe hyponatraemia poses a significant risk for developing ODS [5,8]. Thus, a cautious approach to sodium correction is paramount, guided by frequent monitoring and adherence to established protocols.

Hyponatraemia in the ICU presents multifactorial challenges in diagnosis and management, emphasising the importance of a systematic approach to prevent both the deleterious effects of low sodium levels and the potential harm of rapid correction. However, detailed large datasets from ICU cohorts are lacking, particularly the management strategies utilised when managing patients with severe hyponatraemia. Consequently, this report aims to provide a comprehensive overview of patient characteristics, symptomology, ICU management strategies, the rate of corrections, complications, and outcomes of patients admitted with severe hyponatraemia.

## 2. Materials and Methods

### 2.1. Study Population

This single-centre retrospective observational study of adult patients admitted to the general ICU with severe hyponatraemia (<120 mmol/L) between 23 December 2016 and 8 March 2023 for up to the initial seven days of admission.

### 2.2. Data Collection

The ICU and hospital clinical notes system (MetaVision (iMDsoft, Tel Aviv, Israel)) and CHARTS (custom software for University Hospital Southampton NHS Trust, version 35) were reviewed and yielded all relevant information. Reason for hospital/ICU admission, date of admission/discharge, baseline demographics, past medical history, drug history, and social history were reported. Patients, their families, emergency/ward notes, previous admissions, outpatient records, and general practitioner records are routinely utilised to populate the dataset that was retrospectively reviewed for this study. Alcohol excess disorder was not defined specifically within this report and relied upon patient/family declaration, general practitioner, or previous admission reports.

Symptoms were recorded as confusion, arrhythmia, seizures, weakness, nausea, or vomiting. Admission Glasgow Coma Score (GCS) was also reported. Daily electrolyte panels are sent for all patients with 4–6 hourly arterial blood gas analysis for real-time electrolyte monitoring. Admission types of blood, including thyroid function, random cortisol, serum and urine osmolarities, and urinary sodium, were sent.

Treatments for hyponatraemia included intravenous saline solution infusion, stratified by hypertonic (1.8% or 2.7%) or isotonic (0.9%), hypertonic fluid bolus, sodium chloride oral tablets, vaptans, and fluid restriction. Desmopressin use was also recorded to evaluate overcorrection management.

### 2.3. Outcomes and Definitions

The primary outcome was the prevalence of overcorrection within 48 h of ICU admission (defined as below). The secondary outcomes were hospital survival, ICU survival, ICU length of stay, hospital length of stay, and prevalence of ODS.

Overcorrection was defined according to international clinical practice/consensus guidelines and included an increase in serum sodium of more than (a) 10 mmol/L in the first 24 h, (b) 8 mmol/L in the second 24 h period, or (c) 18 mmol/L within the initial 48 h [5,6].

### 2.4. Ethical Approval

This study is part of a large study investigating the outcome of critically ill patients in the ICU (CRIT-CO) study. This study was sponsored by University Hospital Southampton NHS Foundation Trust (RHM CRI 0370), and ethical approval was obtained from the NHS Health Research Authority, HRA, UK (IRAS 232922) on 26 November 2018. This study was also registered as part of a quality improvement project at University Hospital Southampton NHS Trust (ZAUD 7281). All identifiable patient data are anonymised, and due to the retrospective observational nature of the study, the consent has been waived. This study is compliant with local ethical standards, and no identifiable patient data are presented here.

### 2.5. Statistical Analysis

We tested for normality using the Shapiro–Wilks test, and as our dataset was non-normally distributed, we reported continuous variables as median (inter-quartile range, (IQR)). Baseline characteristics are described by median with IQR for continuous variables and counts with percentages for categorical variables. Kruskal–Wallis test and Fischer’s exact test for continuous and binary outcomes, respectively. Logistic regression models were constructed to analyse predictors of overcorrection at 48 h. Variables were included within the multivariable models based on clinical rationality and an univariable significance threshold of *p* < 0.25. Subsequent backward selection was performed using the Akaike Information Criterion to produce a final model. All analyses were performed using R (version 4.2.2) and regression analysis using the MASS package [9].

## 3. Results

### 3.1. Baseline Characteristics and Overcorrection Incidence

We included 181 patients admitted with severe hyponatraemia (<120 mmol/L) with a median age of 67 years (IQR 52, 77), and 51% were male. The majority (n = 129, 71.3%) were initially admitted to the hospital for clinical presentations not related to hyponatraemia. The comorbidities included hypertension (52%), diabetes mellitus (22%), congestive cardiac failure (15%), ischaemic heart disease (14%), liver cirrhosis (14%), and history of excess alcohol use (28%). The medications on admission included proton pump inhibitors (PPI) (47%), diuretics (33%), antidepressants or antipsychotics (20%), and corticosteroids (14%). The common presenting symptoms were confusion (51%), lethargy (44%), weakness (28%), nausea (27%), and vomiting (24%). Severe symptoms, including seizures and arrythmias, occurred in 14% and 5%, respectively. The median admission sodium, serum osmolality, and urinary sodium were 113 mmol/L (IQR 108, 117), 240 mOsm/kg (IQR 231, 256), and 32 mEq/L (IQR 15, 61), respectively. The treatment offered while in ICU included 0.9% saline infusion (73%), any hypertonic saline infusion (44%), hypertonic saline boluses (29%), and fluid restriction (47%) (Table 1).

We sub-classified the cohort into two groups based on the patients meeting any of the 48 h overcorrection criteria presented in the methods section [5,6]. A breakdown of the number of patients meeting each overcorrection criterion is available in the Appendix A. Overcorrection within 48 h in a combined definition was seen in 62 (34%) patients. Proportionally, the overcorrected group was younger (66 vs. 69 years) and had fewer patients with liver cirrhosis (8.1% vs. 17%). Moreover, on initial presentation, the presence of more severe symptoms, including arrhythmias and seizures, was more common in the overcorrection group (3.4% vs. 8.1% and 10% vs. 23%, respectively). Admission serum sodium and the minimum serum sodium measured within the first week of ICU admission are highly concordant. Furthermore, overcorrected patients exhibit lower admission and minimum sodium values than normal corrected patients (111 mmol/L vs. 114 mmol/L). Overcorrected patients were more likely to receive any hypertonic saline infusion (47% vs. 43%) and were less likely to be treated with fluid restriction or salt tablets (37% vs. 52% and 19% vs. 25%, respectively). We report a low incidence of the use of vaptan medication within our cohort (3.3%) without a significant difference between the normal and overcorrected cohorts.

Figure 1A–C represents the degree of overcorrection seen according to each of the overcorrection criteria outlined in the methods section. A total of 36% of 33 patients overcorrected between 0 and 24 h of ICU admission had a sodium increase between 11 and 12 mmol/L, with the remaining 64% experiencing an increase of >12 mmol/L on day one. A total of 56% of 32 patients overcorrected between 24 and 48 h of ICU admission experienced a sodium increase of 9–10 mmol/L, with the remaining 44% having a rise of >10 mmol/L and 25% of patients experiencing a rise of >15 mmol/L. A total of 50% of 22 patients overcorrected from 0 to 48 h experienced a sodium increase of 19–20 mmol/L, with the remaining having a sodium increase >20 mmol/L.

The sodium correction rate for the entire population over the initial 48 h of admission was a median of 10 mmol/L (IQR: 5–15 mmol/L). For the overcorrection cohort, the median serum sodium increment from 0 to 48 h was 17 mmol/L (IQR: 15–20 mmol/L), and for normal correction 7 mmol/L (IQR: 1–11 mmol/L). From admission to one week, serum sodium levels improved from a median of 113 (IQR 108, 117) to 131 (IQR 126, 135). Stratification into overcorrection or normal correction cohorts is depicted in Figure 2.

### 3.2. Risk Factors for Overcorrection at 48 h

Factors associated with overcorrection at 48 h from univariable analysis were performed (Appendix A). The multivariable model found lower admission sodium value was an independent risk associated with overcorrection at 48 h, whereas liver cirrhosis and use of fluid restriction were independently associated with normal correction within 48 h (Table 2).

### 3.3. Length of Stay

ICU length of stay for the entire cohort was a median of 3 days (IQR 2–4). While there was no difference in the ICU length of stay between these two groups, there was a trend towards shorter overall hospital stay for patients who had overcorrection of sodium levels at 48 h [8 days (IQR 6–17) versus 12 days (IQR 7–22), *p* = 0.084] (Table 3).

### 3.4. Survival and ODS

The overall ICU and hospital survival for the whole cohort was 92% and 87%, respectively, with no difference between overcorrected and normal corrected groups (Table 4 and Figure 3). There was one incidence (0.55%) of ODS in the overcorrection group who had recovered after a period of neurorehabilitation.

## 4. Discussion

In this retrospective cohort study of severe hyponatraemia in the ICU, we reviewed clinical data to determine the incidence and risk factors for overcorrection within 48 h. We also reviewed associations between overcorrection, length of stay, and mortality. We also determined the incidence of ODS in our cohort. We found that 34% of patients met the criteria for overcorrection at 48 h, and lower admission sodium value was an independent risk factor for overcorrection. Furthermore, pre-admission liver cirrhosis and the use of fluid restriction were independent risk factors for normal correction. We identified no statistically significant difference in ICU or hospital length of stay or mortality, although a non-significant trend towards longer hospital length of stay in the normally corrected cohort was noted. ODS was present in 0.55% of severe hyponatremic patients and occurred following overcorrection within 48 h in a patient with risk factors for ODS. To the best of our knowledge, this is the largest UK observation of severe hyponatraemia in ICU patients.

The correction rate of hyponatraemia is based upon several factors, including neurohormonal composition, degree of critical illness, treatment choice, the chronicity of hyponatraemia, and aetiology [10,11,12]. The overcorrection incidence of hyponatraemia is variable within the published literature, occurring in 15–48% of patients [13,14], with the exact figure being dependent upon variable definitions of hyponatraemia and overcorrection. Studies where overcorrection was defined using a higher threshold (>12 mmol/L/day) revealed a lower incidence of overcorrection (14–20.8%) [13,15,16,17], whereas a lower threshold definition (>8 mmol/L/day) revealed a higher incidence (41–44%) [18,19]. Furthermore, where the hyponatraemia definition was set to a higher value (>125 to >130 mmol/L), the incidence of overcorrection was lower (14–19.6%) [16,17]. We found that lower admission sodium was an independent risk factor for overcorrection, a result concordant with other reports [13,16,18], which might explain why studies including more severe hyponatraemic patients report a higher incidence of overcorrection. Study designs similar to this report for both overcorrection and hyponatraemia definitions reveal an incidence between 27.9 and 44.9% [18,20,21], which is in keeping with our findings.

We identified that known liver cirrhosis was independently associated with a normal sodium correction rate. This association is possibly related to the large extravascular fluid volumes that occur in severe liver disease, resulting in a more treatment-resistant state or possibly less use of hypertonic saline with a more fluid-restrictive strategy. George et al. identified that the Charlson Comorbidity Index score (which contains liver disease) was independently associated with overcorrection; however, no breakdown into the score constituents was performed [18].

With regard to treatments utilised for sodium correction, the use of fluid restriction was associated with the normal correction. This is intuitive when one considers that fluid restriction manipulates the ADH response, causing a reduction in water retention in the renal tubules. This process, in comparison to directly administering intravenous sodium chloride, is slower at improving serum sodium concentrations [22,23].

European and American guidelines advocate the use of hypertonic saline infusions/boluses in hyponatraemia with severe symptoms [5,24]. Although efficacious, the effect of hypertonic saline on serum sodium concentration is unpredictable and risks overcorrection [8,25]. We identified that overcorrected patients were more likely to present with severe symptoms (seizures/arrythmias) and were more likely to be treated with 2.7% saline infusions. Concordant with our results, a large cohort study (n = 1490) identified that the use of hypertonic saline was associated with overcorrection in a univariate analysis (*p* < 0.01), although this was not significant in multivariable models [18]. Some studies have demonstrated a clinical benefit of using desmopressin alongside hypertonic saline infusions to facilitate a safe and controlled rise in serum sodium concentration [26,27]; however, a recent randomised controlled trial demonstrated no meaningful differences in sodium correction rate, symptom control, length of stay, ODS, or mortality compared to placebo [28].

ODS is a rare yet devastating consequence of hyponatraemia overcorrection and may result in severe neurological disability and death in 33–55% of cases [29]. The exact incidence of ODS is not known, although observational studies of hyponatraemic patients demonstrate an incidence of 0.2–2% [4,13,18,21,30]. This value is not dissimilar to our findings of 0.55%. Risk factors for ODS include malnutrition, chronic alcoholism and liver cirrhosis, hypokalaemia, hypophosphatemia, and central nervous system hypoxia [5,31,32,33]. The patient we identified who developed ODS within our overcorrection cohort presented with such risk factors.

While our study revealed no difference in ICU length of stay, there was a trend towards prolonged hospital length of stay with normal sodium correction (*p* = 0.087). In a small cohort study of 67 patients, Giordano et al. showed significantly reduced length of stay in the overcorrected cohort vs. normal corrected patients (3.8 days ± 0.4 vs. 10.7 days ± 0.7) [14]. Similarly, Geoghegan et al. evaluated 412 hyponatraemic patients, highlighting a significant increase in hospital length of stay in under-corrected patients (0–5 mmol/L/day) in multivariable analysis [21]. Other studies of a similar cohort size to this report demonstrate non-superiority of normal correction [17,34]. Given the known deleterious effects of prolonged hospital stays both on patients and healthcare providers, such associations should be investigated further, particularly in those with less risk of developing ODS [35].

We demonstrated no significant mortality difference between the over- and normal corrected cohorts in our report, concordant with another study of a similar size [17]. However, a smaller cohort study conducted in the United Kingdom showed a large discrepancy in hospital mortality between overcorrected, normally corrected, and non-responding patients in favour of overcorrection (2.1% vs. 4.5% vs. 20%, respectively) [20]. Larger studies have further demonstrated this mortality benefit [14,18,19,36]. This survival benefit may be explained by less prolonged exposure to detrimentally lower serum sodium concentrations; however, it is important to consider that poorly responsive or unresponsive hyponatraemia may, in fact, be a consequence of a more severe and resistant disease state, which may confound both lengths of stay and mortality outcomes.

It is well established that severe hyponatraemia is associated with poor patient outcomes, regardless of aetiology [4,37]. However, the current guidelines for correction rate in hyponatraemia are designed to prevent neurological complications and ODS [5,24]. Given the presented data and summary of the literature, such complications are rare, associated with clear risk factors, and, in some reports, are not associated with overcorrection at all [19]. In fact, reduced mortality, improved survival, and reduced length of stay should call into question whether overcorrection is preferable in selective groups of patients with minimal risk of ODS. Perhaps a more personalised target should be applied. Ayus and Moritz (2023) suggested that the current sodium correction targets are restrictive and prohibit the use of effective hypertonic saline treatment. They, therefore, propose a more liberal target of 15–20 mmol/L/day; however, at present, no research has been conducted to prove both the efficacy and safety of this target [38]. Sterns et al., in a recent report, reiterate the potential risk of ODS in severe hyponatraemic patients with overcorrection. The authors note the possibility of underreporting given that magnetic resonance imaging (MRI) is not 100% sensitive for ODS diagnosis and correct coding is not always achieved, meaning true ODS patients are potentially missed in retrospective analyses [39]. Furthermore, in reports where ODS is identified in patients without an observed overcorrection, there is a risk that they may have already experienced an overcorrection event prior to hospital attendance. Overall, this highlights the need for a randomised trial evaluating different sodium correction targets before different treatment targets are advocated.

The main strength of our study was its comprehensive patient capture over six years of clinical practice. The patients are representative of hyponatraemia patients within an ICU setting, and the treatment options utilised are concordant with current applicable guidelines. We also utilised an overcorrection definition that was in accordance with European guidelines and inclusive of all patients overcorrected within 48 h. However, there are several limitations of this observational study that are worthy of note. This is pragmatic, real-world clinical practice data. Although treatment guidelines both locally and nationally were followed within this cohort, with a small population size from only a single centre, there is a risk of incongruency between the local practice used in our unit and elsewhere. Additionally, given the rarity of ODS within this population, it is difficult to perform any meaningful statistical analysis. This study was performed as a retrospective data analysis, meaning key clinical information has not been captured, including volume status on arrival to the hospital/ICU, a clear working diagnosis for hyponatraemia, and acute or chronic hyponatraemia status. There were additionally some missing data for serum and urine osmolarities, urine sodium, cortisol, and TSH levels; however, this has been documented clearly in the Appendix A.

## 5. Conclusions

This retrospective single-centre cohort study has identified that overcorrection of severe hyponatraemia is common and is more likely to occur in younger patients without liver cirrhosis, presenting with a lower admission serum sodium concentration and severe symptoms, and treated with 2.7% saline infusions. However, in multivariate analysis, only the admission of lower sodium levels was associated with overcorrection. ODS is rare and occurs in less than 1% of our cohort. Correction rates for hyponatraemia are designed to prevent ODS; however, with well-defined risk factors for ODS and the risk of prolonged length of stay and mortality with inadequate hyponatraemia management, future research should be directed towards determining whether a more liberal target for select severe hyponatraemia patients is safe and effective.

## Figures and Tables

**Figure 1 medicina-60-01412-f001:**
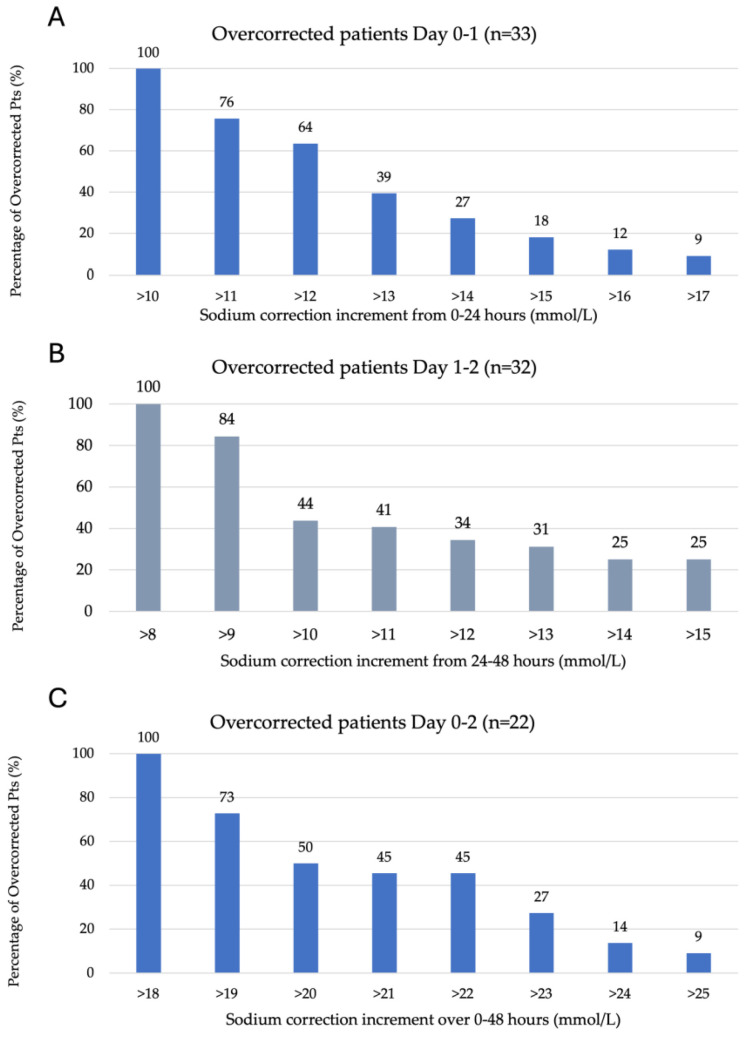
Serum sodium kinetics over the initial 48 h of ICU admission. (**A**) Overcorrection by day one (0–24 h), defined as >10 mmol/L (n = 33). (**B**) Overcorrection on day two (24–48 h) is defined as >8 mmol/L (n = 32). (**C**) Overcorrection from admission to day two (0–48 h) is defined as >18 mmol/L (n = 22).

**Figure 2 medicina-60-01412-f002:**
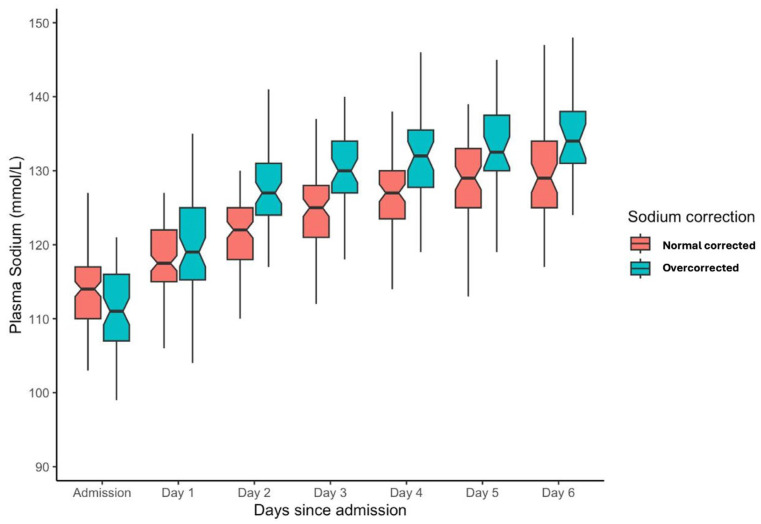
Box plot for serum sodium level from admission for the initial week of ICU admission stratified into overcorrection and normal correction cohorts.

**Figure 3 medicina-60-01412-f003:**
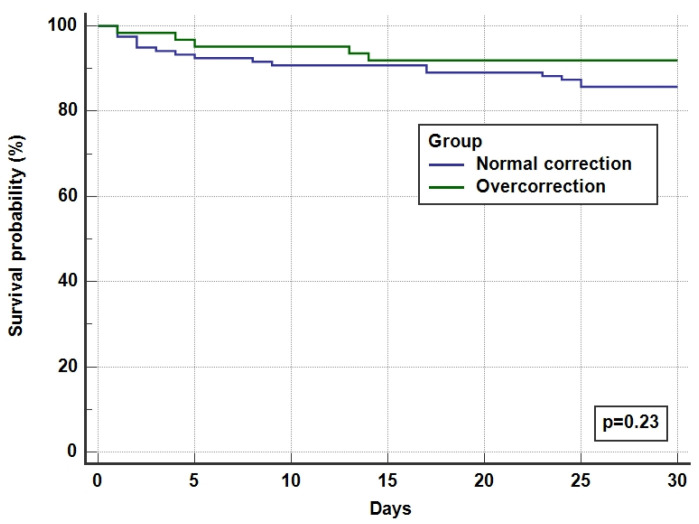
Kaplan Meier survival curves for 30-day hospital survival.

**Table 1 medicina-60-01412-t001:** Baseline characteristics table.

Characteristic	Total (n = 181) ^1^	Normal Correction (n = 119) ^1^	Overcorrected (n = 62) ^1^
Age	67 (52, 77)	69 (54, 78)	66 (51, 76)
Gender (Male)	93 (51%)	61 (51%)	32 (52%)
Hypertension	94 (52%)	62 (52%)	32 (52%)
Alcohol Excess *	51 (28%)	35 (29%)	16 (26%)
Type 2 Diabetes	33 (18%)	24 (20%)	9 (15%)
Congestive cardiac failure	27 (15%)	18 (15%)	9 (15%)
Ischaemic heart disease	26 (14%)	15 (13%)	11 (18%)
Liver Cirrhosis	25 (14%)	20 (17%)	5 (8.1%)
Chronic steroid use	13 (7.2%)	8 (6.7%)	5 (8.1%)
Chronic kidney disease	12 (6.6%)	7 (5.9%)	5 (8.1%)
Type 1 Diabetes	7 (3.9%)	3 (2.5%)	4 (6.5%)
Proton pump inhibitors (PPI)	85 (47%)	52 (44%)	33 (53%)
Diuretics (any)	59 (33%)	43 (36%)	16 (26%)
Antidepressant/antipsychotics	37 (20%)	24 (20%)	13 (21%)
Loop diuretic	30 (17%)	22 (18%)	8 (13%)
Steroids	25 (14%)	16 (13%)	9 (15%)
Thiazide diuretic	25 (14%)	17 (14%)	8 (13%)
Potassium-sparing diuretic	23 (13%)	16 (13%)	7 (11%)
Amiodarone	2 (1.1%)	1 (0.8%)	1 (1.6%)
Carbamazepine	1 (0.6%)	0 (0%)	1 (1.6%)
Admission GCS	15.0 (14.0, 15.0)	15.0 (14.0, 15.0)	15.0 (14.0, 15.0)
Confusion	93 (51%)	58 (49%)	35 (56%)
Lethargy	79 (44%)	47 (39%)	32 (52%)
Weakness	50 (28%)	29 (24%)	21 (34%)
Nausea	48 (27%)	30 (25%)	18 (29%)
Vomiting	44 (24%)	26 (22%)	18 (29%)
Seizures	26 (14%)	12 (10%)	14 (23%)
Neurological insult *	13 (7.2%)	11 (9.3%)	2 (3.2%)
Arrhythmias	9 (5.0%)	4 (3.4%)	5 (8.1%)
Admission Na (mmol/L)	113 (108, 117)	114 (110, 117)	111 (107, 116)
Baseline Creatinine (μmol/L) *	60 (46, 82)	62 (46, 86)	56 (45, 68)
Cortisol (nmol/L) *	537 (390, 815)	547 (403, 815)	516 (321, 778)
Lowest Na in the first week (mmol/L)	113 (108, 116)	114 (110, 116)	111 (107, 116)
Serum osmolarity (mOsm/kg) *	240 (231, 256)	240 (233, 256)	242 (226, 256)
T4 (pmol/L) *	15.0 (12.4, 19.2)	15.1 (12.0, 19.2)	14.8 (13.2, 20.2)
TSH (miU/L) *	1.36 (0.69, 2.50)	1.48 (0.69, 2.66)	1.18 (0.69, 2.05)
Urine sodium (mEq/L) *	32 (15, 61)	31 (16, 57)	34 (15, 62)
0.9% saline infusion	132 (73%)	81 (68%)	51 (82%)
1.8% saline infusion	65 (36%)	45 (38%)	20 (32%)
2.7% saline infusion	24 (13%)	12 (10%)	12 (19%)
Any hypertonic saline infusion (alone or in combination)	80 (44%)	51 (43%)	29 (47%)
Desmopressin	25 (14%)	18 (15%)	7 (11%)
Fluid restriction	85 (47%)	62 (52%)	23 (37%)
Hypertonic saline bolus	52 (29%)	33 (28%)	19 (31%)
Salt tablets	42 (23%)	30 (25%)	12 (19%)
Vaptans	6 (3.3%)	4 (3.4%)	2 (3.2%)

^1^ Median (IQR); n (%). Neurological insult is defined as admission for either transient ischaemic attack, cerebrovascular event, brain tumour, or head trauma. NaCl: sodium chloride, TSH: thyroid stimulating hormone, T4: free thyroxine hormone. * Missing data within groups. Available in Appendix A.

**Table 2 medicina-60-01412-t002:** Multivariable analysis was deduced from the backward selection using the Akaike Information Criterion method.

Characteristic	Odds Ratio	95% Confidence Interval	*p*-Value
Age	0.98	0.96, 1.00	0.083
Admission Sodium level value	0.91	0.86, 0.96	0.001
Liver cirrhosis	0.30	0.09, 0.86	0.035
Fluid restriction	0.46	0.23, 0.90	0.025

**Table 3 medicina-60-01412-t003:** ICU and hospital length of stay.

Length of Stay	Total ^1^	Normal Correction ^1^	Overcorrection ^1^	*p*-Value ^2^
ICU (days)	3.0 (2, 4)	3 (2, 4)	3.0 (2, 5)	0.15
Hospital (days)	11 (6, 21)	12 (7, 22)	8 (6, 17)	0.084

^1^ Median (IQR). ^2^ Wilcoxon rank sum test.

**Table 4 medicina-60-01412-t004:** ICU and hospital survival.

Survival	Total ^1^	Normal Correction ^1^	Overcorrection ^1^	*p*-Value ^2^
Discharged Alive ICU	166 (92%)	108 (91%)	58 (95%)	0.4
Discharge Alive Hospital	158 (87%)	103 (87%)	55 (89%)	0.7

^1^ n (%). ^2^ Fisher’s exact test.

## Data Availability

The study data are available upon request.

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
