# Peer review of "Intensive Care Management of Severe Hyponatraemia—An Observational Study"

_medicina, 2024, doi:10.3390/medicina60091412_

Round 1
Reviewer 1 Report
Comments and Suggestions for Authors
- The authors must better describe how to decrease SNa in overcorrected patients (combining water intake with dDAVP : one liter of water with dDAVP
will decrease SNa by 4-6 mEq/L (depending of BW etc)
Vaptans are not proposed to treat symptomatic hyponatremia (see Hypertonic saline,isotonic saline,water restriction,etc...Expert Review of endocrinology &Metabolism 2020 )
Over corrected patients must be decreased.1 liter of water combine with dDAVPwill decrease SNa by 4_6 mEq/L(depending of body weight)
Some intensive care units use urea to treat severe hyponatremia (critical care)
A recent article by Sterns et al (CJASN 2024)contest the limits proposed by AYUS (to mention)
Author Response
We are sincerely grateful for the reviewer's comments. Please see our responses below.
- The authors must better describe how to decrease SNa in overcorrected patients (combining water intake with dDAVP : one liter of water with dDAVP will decrease SNa by 4-6 mEq/L (depending of BW etc).
Response: Thank you for this comment. We have now included this in the introduction section (line 48-52), highlighting this point.
- Vaptans are not proposed to treat symptomatic hyponatremia (see Hypertonic saline,isotonic saline,water restriction,etc...Expert Review of endocrinology &Metabolism 2020 )
Response: Thank you for highlighting this and we agree with the reviewer and removed this from the statement in the introduction (line 47). This is consistent with our findings that only a small proportion (<4%) use in our intensive care unit and highlighted in lines 148-150 why we included this within our manuscript.
- Over corrected patients must be decreased.1 liter of water combine with dDAVPwill decrease SNa by 4_6 mEq/L(depending of body weight).
Response: Thank you for this point. We have commented on the use of desmopressin with and without the use of water in the introduction (lines 48-52). We reference Spasovski et al (2014) who highlight that recommendation but advise expert endocrine opinion before commencing desmopressin treatment and further research into the efficacy of such a strategy.
- Some intensive care units use urea to treat severe hyponatremia (critical care)
Response: We recognise the potential efficacy of urea in hyponatraemia. However, we could not comment on the use of this substance within our cohort because it is not standard practice in our unit.
- A recent article by Sterns et al (CJASN 2024) contest the limits proposed by AYUS (to mention)
Response: Thank you for the reviewer’s comments. We have included a sentence to reflect the reviewer’s comments within the discussion, referencing the points described by Sterns et al to maintain the current targets of Na correction (see lines 307-315).
Reviewer 2 Report
Comments and Suggestions for Authors
Dear author
Written informed consent is needed
In figure 3 why only 30-day hospital survival is presented?
What is the novelty of work regrading to previous publication
- Sim JK, Ko RE, Na SJ, Suh GY, Jeon K. Intensive care unit-acquired hyponatremia in critically ill medical patients. Journal of Translational Medicine. 2020 Jul 2;18(1):268.
- Kinoshita T, Mlodzinski E, Xiao Q, Sherak R, Raines NH, Celi LA. Effects of correction rate for severe hyponatremia in the intensive care unit on patient outcomes. Journal of Critical Care. 2023 Oct 1;77:154325.
- Sterns RH, Rondon-Berrios H, Adrogue HJ, Berl T, Burst V, Cohen DM, Christ-Crain M, Cuesta M, Decaux G, Emmett M, Garrahy A. Treatment guidelines for hyponatremia: stay the course. Clinical Journal of the American Society of Nephrology. 2024 Jan 1;19(1):129-35.
Comments on the Quality of English LanguageMinor editing of English language required
The cohesion and coherence should be increased
Author Response
We are sincerely grateful for the reviewers comments and please see our responses below.
- Written informed consent is needed
Response: Thank you for this comment. This study is conducted as part of the evaluating outcomes in critically ill patients (CRIT-CO). This is a retrospective observational study designed to evaluate common presentations and outcomes in ICU. We have the appropriate ethics approval (HRA IRAS ID 232922) for the study. The consent is waived due to the nature of the study and no identifiable patient data is presented here.
- In figure 3 why only 30-day hospital survival is presented?
Response: Mortality beyond 30 days is more likely due to other factors beyond the initial presentation and management of hyponatraemia. Most of our patients were discharged from ICU by this point. Moreover, given that sodium concentrations within our cohort of severe hyponatraemic patients demonstrated an improvement to safe levels within 1 week of admission (median 113 (IQR 108, 117) to 131 (IQR 126, 135), we believe that 30-day mortality provides the best means of causality to hyponatraemia within the limitations of our dataset.
- What is the novelty of work regrading to previous publication
- Sim JK, Ko RE, Na SJ, Suh GY, Jeon K. Intensive care unit-acquired hyponatremia in critically ill medical patients. Journal of Translational Medicine. 2020 Jul 2;18(1):268.
- Kinoshita T, Mlodzinski E, Xiao Q, Sherak R, Raines NH, Celi LA. Effects of correction rate for severe hyponatremia in the intensive care unit on patient outcomes. Journal of Critical Care. 2023 Oct 1;77:154325.
- Sterns RH, Rondon-Berrios H, Adrogue HJ, Berl T, Burst V, Cohen DM, Christ-Crain M, Cuesta M, Decaux G, Emmett M, Garrahy A. Treatment guidelines for hyponatremia: stay the course. Clinical Journal of the American Society of Nephrology. 2024 Jan 1;19(1):129-35.
Response: This study represents the largest retrospective cohort of severe hyponatraemia patients (<120mmol/L) from the United Kingdom. Furthermore, the definition of overcorrection encompassing the initial 48 hours of admission is concordant with the current available guidelines (both European and American) for management of severe hyponatraemia. We produced findings that are concordant with larger international studies, indicating that our sample acts as a good estimation of the true state of severe hyponatraemia managed in the intensive care unit.